# Excitatory rubral cells encode the acquisition of novel complex motor tasks

Giorgio Rizzi[1], Mustafa Coban[1] & Kelly R. Tan[1]

The red nucleus (RN) is required for limb control, specifically fine motor coordination. There is some evidence for a role of the RN in reaching and grasping, mainly from lesion studies, but results so far have been inconsistent. In addition, the role of RN neurons in such learned motor functions at the level of synaptic transmission has been largely neglected. Here, we show that Vglut2-expressing RN neurons undergo plastic events and encode the optimization of fine movements. RN light-ablation severely impairs reaching and grasping functions while sparing general locomotion. We identify a neuronal population co-expressing Vglut2, PV and C1QL2, which specifically undergoes training-dependent plasticity. Selective chemo-genetic inhibition of these neurons perturbs reaching and grasping skills. Our study highlights the role of the Vglut2-positive rubral population in complex fine motor tasks, with its related plasticity representing an important starting point for the investigation of mechanistic substrates of fine motor coordination training.

[1] Biozentrum, University of Basel, Klingelbergstrasse 50/70, 4056 Basel, Switzerland. Correspondence and requests for materials should be addressed to K.R.T. (email: kelly.tan@unibas.ch)

The red nucleus (RN) is a large area located within the midbrain tegmentum. It is part of the motor circuit involving the motor cortex[1], the posterior thalamus[2], the cerebellum[3,4] and the spinal cord[5]. Its roles in limb control and fine motor coordination[6] are well accepted. However, the studies elucidating the RN control of reaching movements have mainly used excitotoxic chemical approaches[7–9]. A particular resiliency of rubral cells has made consistent lesions hard to achieve. Furthermore, variability capture (standard vs high-speed), as well as levels decomposition of reaching movements (approach, extension, supination pronation, arpeggio etc…), have led to different interpretations of the role of the RN in fine motor skills[9–11].

An additional level of complexity depends on the presence of cellular heterogeneity in the RN. Staining studies performed on different species have shown that neurons of the RN fall into three general categories based on size: small, medium and large. The largest cells are generally found within the caudal part, while in the middle and rostral parts of the nucleus, populations of medium-sized and small cells appear[12,13]. These experimental observations are in line with the concept of the large RN magnocellular neurons (RNm) encompassing the caudal portion and the RN parvocellular neurons (RNp) making up the smaller rostral area[14]. The RNm have been reported to be part of the rubrospinal pathway whereas the RNp are part of the motor loop involving the inferior olive and cerebellum[15].

Aside from lesioning studies, in vivo electrophysiological recordings of RN neurons and electromyographic activity during limb activity in non-human primates have also suggested the RN as a major brain structure involved in the execution of fine movements. Specifically, the discharge of individual RNm was reported to correlate with kinematic variables of limb movements such as movement velocity[16,17] and also correlate with digits and wrist extensor and shoulder flexor muscles[18,19]. Furthermore, in a reach-to-grasp task, RNm were proposed to contribute to the control of hand pre-shaping[20].

Synaptic plasticity is postulated as the main mechanism to encode new information in the brain and is largely investigated in other brain regions but to a much lesser extend in the RN. Anatomical plasticity has been observed at the cortico-rubral synapses. Classical conditioning in cats has shown that after pairing cerebral peduncle stimulation with an electric shock to the forelimb skin, cats would present a forearm retraction in response to cerebral peduncle stimulation, a motor action previously not present[21]. Under such experimental paradigm, electron microscopy analysis reported an increased number of synaptic contacts received by proximal dendrites and soma of RN neurons[22]. In addition, in vivo recordings of RN cells highlighted an increase in firing probability in conditioned but not in control animals[23]. This effect was only observed with cerebral peduncle stimulation but not with that of the interpositus nucleus stimulation acting as the conditioned stimulation[24].

Here, we augment the knowledge surrounding the role of the RN in motor functions from fine movement skills including reaching and grasping, to gross movements as observed in spontaneous locomotion. Specifically, we observe that a selective sub-population of the RN is implicated in the encoding of complex and new fine motor skills and does so via functional changes in synaptic plasticity. We show with a novel light lesioning approach that the RN plays an important role in the control of finer aspects of motor output. We further elucidate that the neuronal population co-expressing Vglut2 and PV have a unique electrophysiological profile, anatomical output targets and underlies the RN mediated fine motor skills. Precisely, we show that this specific neuronal population undergoes excitatory synaptic plasticity during the learning of reaching movements and their selective inhibition impairs reaching and grasping skills.

## Results

### RN lesion impairs reaching and grasping, sparing locomotion.
We first wanted to establish to what extent ablation of the RN would impair motor performance. We designed a multi-step paradigm where the performance of mice on three different motor tasks requiring respective increased dexterity would be evaluated before and after rubral ablation. To this end, mice underwent an open field spontaneous locomotion test (OFT, Supplementary Movie 1), suspended grid test (SGT[25,26], Supplementary Movie 2) and a single pellet reaching task (SPRT[8,27], Supplementary Movie 3). Mice were kept under a food restriction regimen starting a week before the beginning of their training in the SPRT and once they reached a stable performance across three consecutive days, their motor dexterity was evaluated across all three behavioural tests (Fig. 1a). Subsequently, the RN was ablated using high power blue light, and their performance was re-evaluated 48-h post lesion across all three tests as well the performance on the SPRT and the SGT one week later (Fig. 1a). A separate cohort of animals was previously used to validate the ablation protocol and revealed a loss of approximately 90% of neurons (Fig. 1b). The spontaneous locomotion in the OFT was decomposed into locomotion bouts[28]. Although a general decrease in all the OFT locomotion parameters was observed, none of them were statistically significant (Fig. 1c–g) suggesting that the bilateral ablation of the RN has a minimal impact on general locomotion.

Upon evaluation of the performance in the SGT, the locomotion accuracy was significantly impaired across both the right and the left hindlimbs (HLr, HLl) as well as the forelimbs (FLr, FLl) 48 h after the lesion and a partial recovery was observed one week later as suggested by the failure rate and the time to complete the initial 50 steps for each limb (Fig. 1h, i). In order to eliminate a learning curve confound in the interpretation of these results, a naive group of mice was tested on the SGT following the same time points but without any lesion. No difference was observed in the performance of these mice across the three-time points (Supplementary Fig. 1).

The highest degree of motor impairment due to RN lesion was observed in the SPRT where the failure rate was significantly higher 48 h post injury and remained so 1 week later despite the time needed to complete the session and the number of overall reaching attempts going back to baseline levels (Fig. 1j–o), suggesting that proper RN function is necessary for appropriate motor coordination and furthermore, tasks requiring a higher degree of fine control will be more strongly affected by RN ablation.

### Motor training induces plasticity on excitatory synapses.
Given these results, we hypothesized that the acquisition and optimization of a complex motor program during the SPRT training period would be represented within the RN and retained via the expression of long-lasting plastic changes at the synapse level. To test this hypothesis a new cohort of mice underwent the same food restriction protocol and was either trained in the SPRT until reaching the stability criteria (Fig. 2a–c) or was exposed to the SPRT box without performing any reaching movement training. In vitro electrophysiological recordings were then performed on brain slices containing the RN from either trained or control animals. We observed no difference in the AMPA/NMDA ratio of RN cells between trained and control mice (Fig. 2d–f). However, the training-induced a switch from rectifying to linear synapses[29–31] in RN cells with a large capacitance (Fig. 2g–i) confirming that motor training of a skilled complex movement induces plastic changes onto RN neurons and suggesting that a subset of RN cells might primarily govern this particular motor feature.

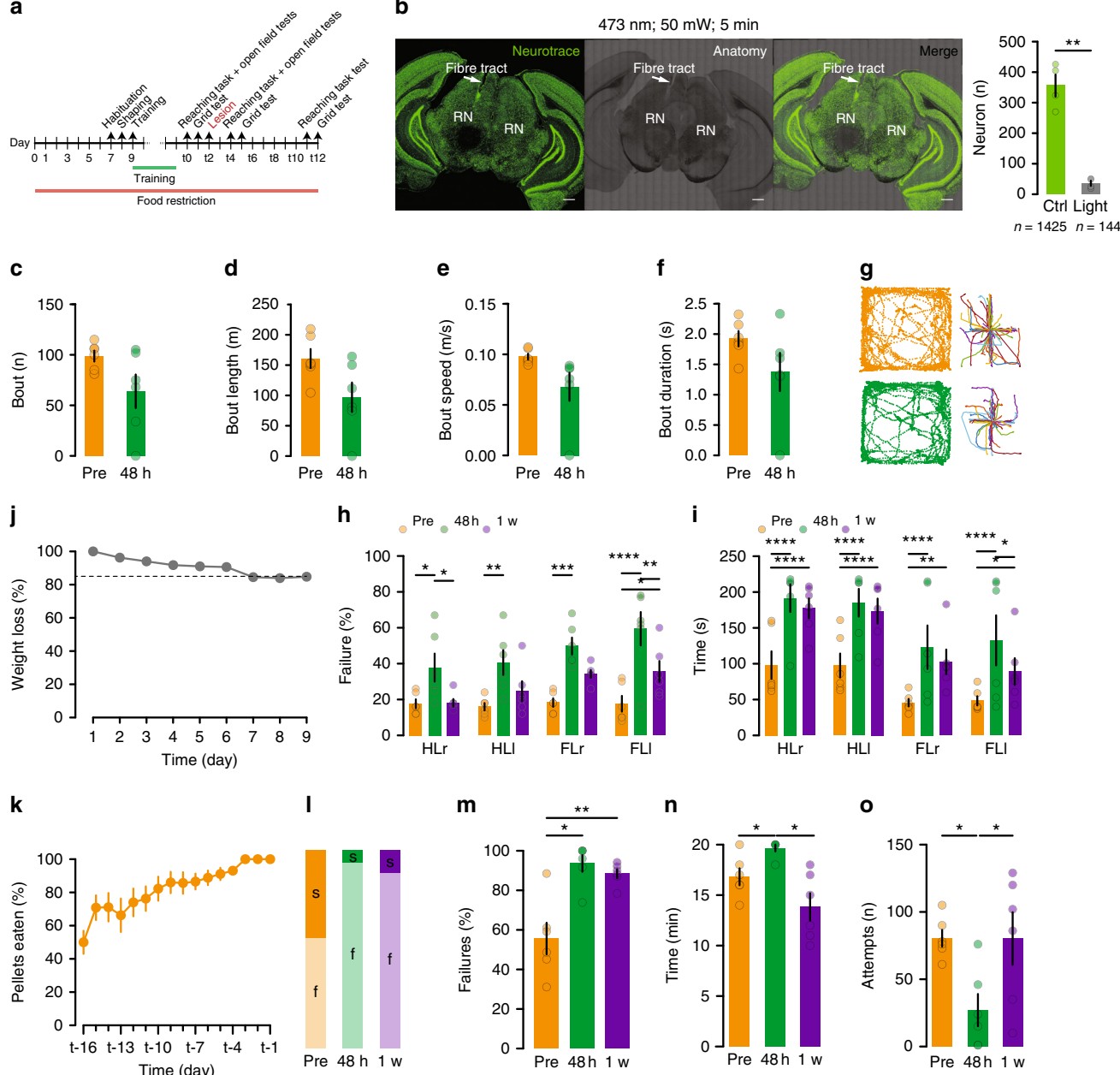

**Fig. 1** RN ablation impairs reaching and grasping, sparing locomotion. **a** Experimental design. **b** Neurotrace staining and lesion protocol validation and quantification $t(3) = 8.36$, $P = 0.0036$, $n = 4$ mice (Scale bar, 500 μm). **c** Number, **d** length, **e** speed and **f** duration of bouts during locomotion in the OFT ($n = 6$ mice). **g** Example track plots and bout decomposition. **h** Failure for each limb in the EGT, 2WRM-ANOVA $F(2,40) = 36.21$, $P < 0.0001$ (HL: hindlimb, FL: forelimb, r: right, l: left). **i** Completion time, 2WRM-ANOVA $F(2,40) = 23.81$, $P < 0.0001$. **j** Weight loss stabilization at 85% body weight during the food restriction period. **k** Learning curve as pellets consumed in the SPRT. **l** Distribution of success and failure across the 3-time points. **m** Failure rate, 1WRM-ANOVA $F(1.3,6.5) = 17.83$, $P = 0.0035$. **n** Completion time, 1WRM-ANOVA $F(1.2,5.9) = 8.43$, $P = 0.0251$. **o** Number of attempts, 1WRM-ANOVA $F(1.6,7.8) = 8.79$, $P = 0.0127$. All data are reported as mean and SEM. Pre-lesion performance are depicted in orange (Pre), performance 48 h after lesion are shown in green and performance a week after lesion (1 w) are displayed in purple

**Vglut2- cells co-express PV and C1QL2.** We took a step towards characterizing the genetic identity of neurons in the RN and fluorescent in-situ hybridization (FISH) revealed a rather minimal presence of inhibitory Vgat-positive cells along with a predominant population of excitatory Vglut2-positive cells that interestingly, co-express the calcium-binding protein parvalbumin (PV) to a large extent (Fig. 3a, b). This finding adds to an exciting and emerging research field reporting the presence of PV in excitatory neurons in sub-cortical areas[32]. This pattern was also observed when probing the expression of Vgat and Vglut2

against C1QL2, a complement protein that has previously been reported as a specific marker for RN neurons[33] (Fig. 3c, d, Supplementary Fig. 2). Finally, we confirmed the above result with a triple co-localization FISH assay that demonstrated the abundant presence of Vglut2, PV and C1QL2 in the RN region (Fig. 3e, f).

**Vglut2- cells exhibit specific intrinsic properties.** Following the FISH evidence suggesting that the excitatory Vglut2-positive cells are the main population in the RN, we performed in vitro

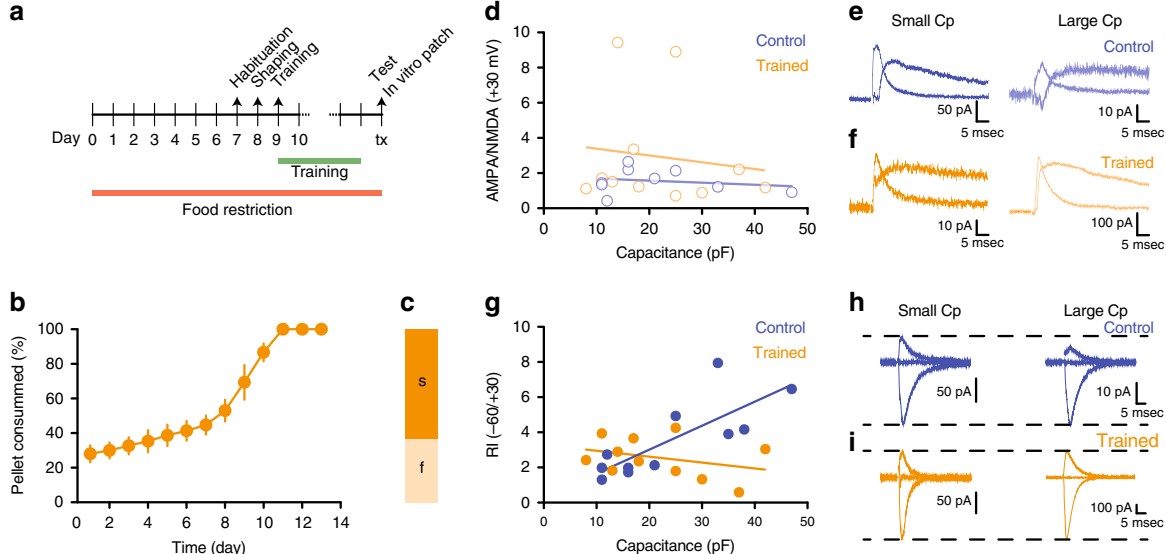

**Fig. 2** Motor training induces plasticity on excitatory synapses. **a** Experimental design for plasticity experiments after SPRT. **b** Pellets consumed during the training. **c** Performance on the test day. **d–f** AMPA/NMDA ratio vs cell capacitance ($n = 9–11$ cells/group) (**d**), with corresponding example traces in **e** and **f**. **g–i** Rectification index vs cell capacitance ($n = 11$ cells/group) (**g**) and representative traces in **h** and **i**. All data are reported as mean and SEM. Results obtained from control mice are displayed in blue whereas data obtained from trained mice are shown in orange

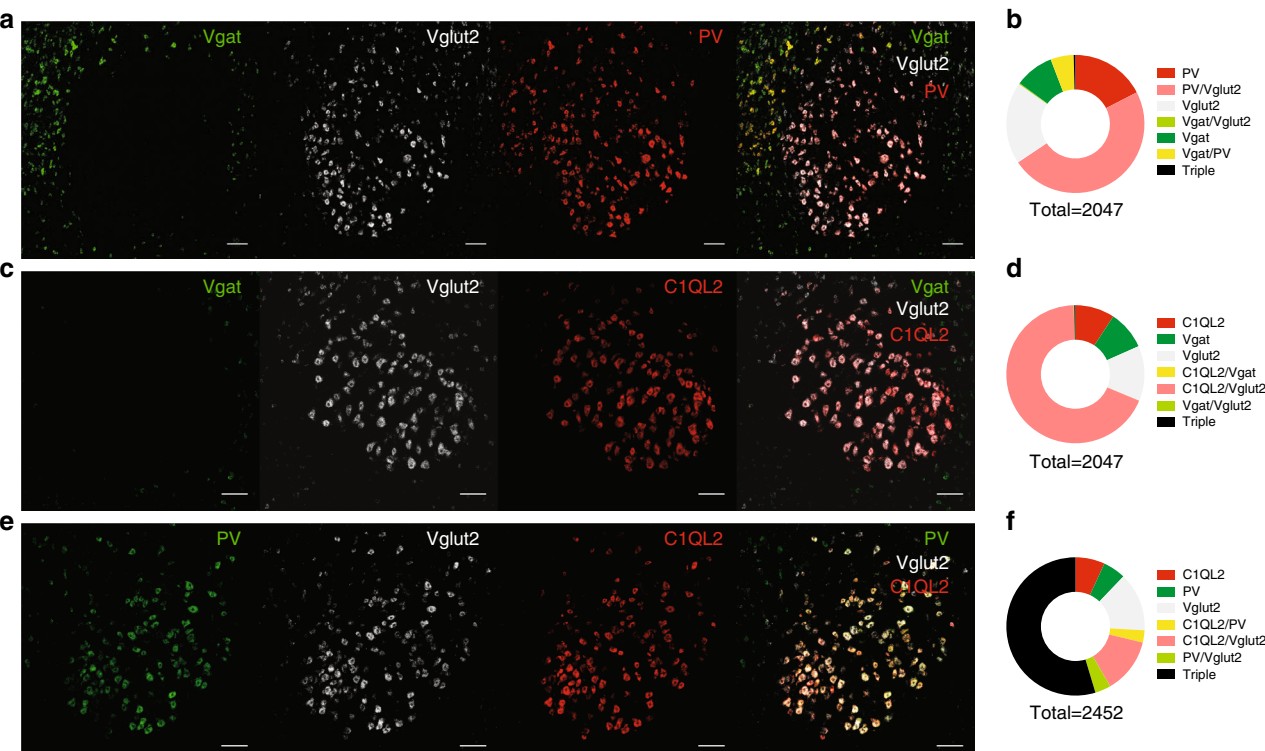

**Fig. 3** Vglut2-positive cells co-express PV and C1QL2. **a**, **b** Confocal images of Vgat, Vglut2 and PV mRNA (**a**) with corresponding quantification (scale bar, 100 μm) in **b**. **c**, **d** FISH for Vgat, Vglut2 and C1QL2 (**c**) and quantification in **d**. **e**, **f** mRNA for PV, Vglut2 and C1QL2 (**e**) with quantification in **f**. ($n = 3$ mice/condition)

whole-cell patch-clamp experiments in RN-containing slices from Vglut2-CRE mice[34] injected with a viral vector coding the expression of channelrhodopsin—2 (AAV5-EF1α-DIO-ChR2-EYFP) (Fig. 4a), used to photo-tag Vglut2-positive neurons (Fig. 4b). Overall RN Vglut2-positive cells were able to sustain a much higher firing frequency in response to larger current injections, displayed a larger capacitance, higher rheobase, saturating currents and maximum firing rate (Fig. 4c–h), a

narrower action potential (Fig. 4i) and had similar resting membrane potentials to Vglut2-negative cells (Fig. 4j).

**Vglut2- cells exhibit specific output targets**. To further characterize the Vglut2-positive neurons, we assessed their downstream projections and injected Vglut2-CRE mice in the RN with a CRE-dependent viral vector coding a GFP tagged form of synaptophysin together with cytosolic tdTomato (AAV5-phSyn1

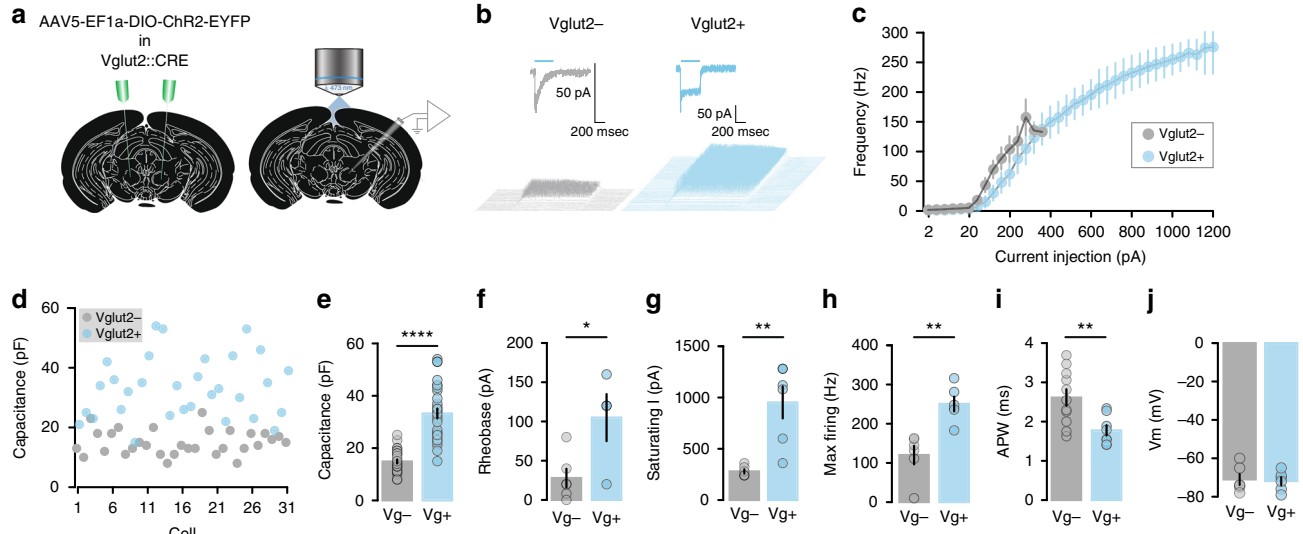

**Fig. 4** Vglut2-positive cells exhibit specific intrinsic properties. **a** Schematic of viral injection of ChR2-eYFP in Vglut2-CRE RN and in vitro whole-cell patch-clamp recordings. **b** Light induced-currents and firing activity in identified cells. **c** Input-output curves ($n = 6$ cells/group). **d** Capacitance distribution of Vglut2-positive and Vglut2-negative cells ($n = 31$ cells/group). **e** Capacitance, $t(60) = 9.12$, $P < 0.0001$. **f** Rheobase, $t(8) = 2.77$, $P = 0.0244$ ($n = 8$–12 cells/group). **g** Saturating current, $t(10) = 4.27$, $P = 0.0016$. **h** Maximal firing, $t(10) = 4.38$, $P = 0.0014$. **i** Action potential width, $t(17) = 3.01$, $P = 0.0067$. **j** Resting membrane potential. ($n = 6$ cells/group). All data are reported as mean and SEM. Data obtained from photo-identified Vglut2-positive neurons are shown in blue (Vg+) whereas those light-insensitive Vglut2-negative neurons are displayed in grey (Vg−)

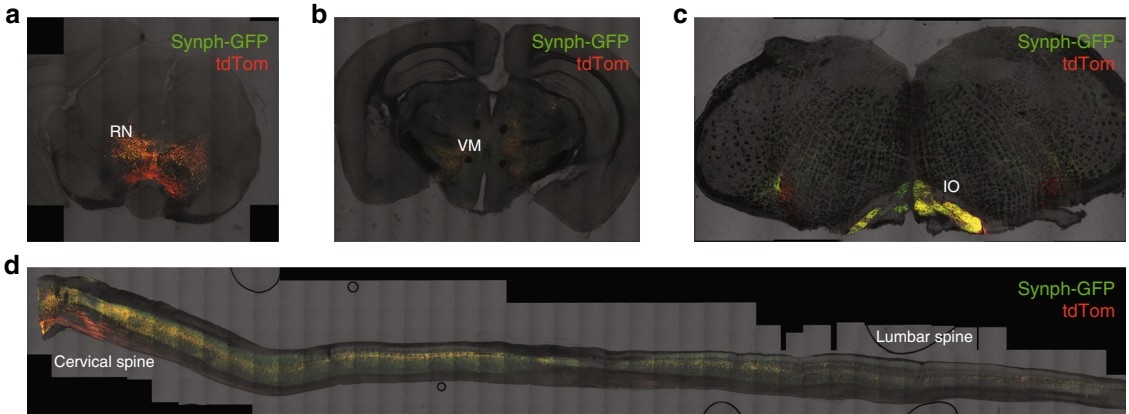

**Fig. 5** Vglut2-positive cells project to thalamic, inferior olive and spinal cord areas. **a** Example images of synaptophysin-GFP-td-Tomato expression in the RN of Vglut2-CRE mice ($n = 4$ mice). **b** Coronal view of Vglut2- RN innervation in the thalamus. **c** Coronal section of the brainstem. **d** Sagittal view of the spinal cord

(S)-FLEX-tdTomato-T2A-SypEGFP-WPRE) (Fig. 5a). In all the injected animals we observed robust innervations in the ventromedial portion of the Thalamus, the Inferior Olive and Spinal Cord (Fig. 5b–d), consistent with the previously reported rubrospinal pathway[35–37].

Considering that in vitro characterization of RN neurons uncovered Vglut2-positive cells as having an overall larger capacitance compared to negative cells (Fig. 4d, e), and that blind evaluation of plastic changes in the RN suggest that large capacitance cells undergo a switch in rectification index after motor training (Fig. 2g), we hypothesized that Vglut2-positive neurons in the RN are necessary for the acquisition and retention of skilled motor movements in mice.

**Vglut2- cells undergo training-dependent plasticity.** To confirm that Vglut2-positive neurons are undergoing plasticity, encoding the acquisition of the reaching task, we first performed in vitro whole-cell patch-clamp recordings in Vglut2-CRE mice injected

with DIO-ChR2 in the RN that went through the motor training necessary to reach a stable performance in the SPRT (Fig. 6a, b). As previously observed, we found no difference in the AMPA/ NMDA ratio between trained vs. control animals neither in Vglut2-positive nor in Vglu2-negative cells (Supplementary Fig. 3). In contrary, and as we predicted, we observed a switch from rectifying to linear synapses only in Vglut2-positive cells from trained mice (Fig. 6c–g). Furthermore, the motor training induced a subunit composition switch from GluA2-lacking $Ca^{2+}$ permeable AMPARs to GluA2-containing $Ca^{2+}$ impermeable AMPARs, as evidenced by the ability of NASPM (a selective $Ca^{2+}$ permeable AMPAR antagonist[38]) to partially block EPSCs and lower the RI in control but not in trained animals (Fig. 6h–j). In a separate set of experiments, we tested if inhibitory transmission was also altered following motor training, but found no differences between control and trained mice (Supplementary Fig. 4).

These data provide previously unreported evidence that a genetically specific subset of RN neurons (Vglut2) undergoes

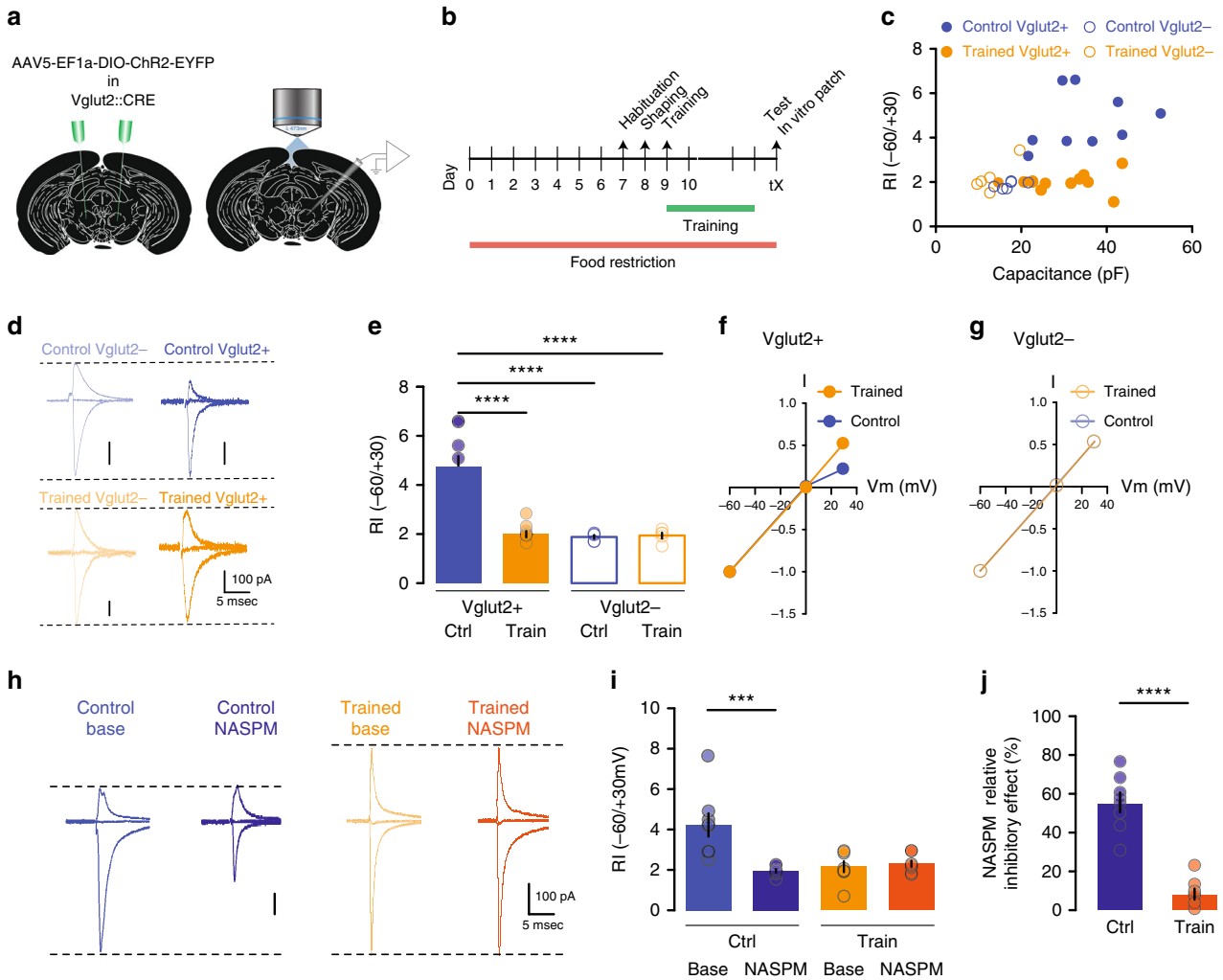

**Fig. 6 Vglut2-positive neurons undergo training-dependent plasticity. a** Schematic of ChR2-eYFP injection in the RN of Vglut2-CRE mice and patch-clamp recording. **b** Timeline. **c, d** Rectification index vs cell capacitance (**c**) with example traces ($n = 6$–$11$ cells/group) in **d**. **e–g** Rectification index (**e**), 2W-ANOVA $F_{(1,26)} = 22.72$, $P < 0.0001$ and corresponding I/V curves in **f** and **g** ($n = 5$–$11$ cells/group). **h** Example EPSC traces before and after bath-application of NASPM ($n = 8$ cells/group). **i** Effect of NASPM on Rectification index. **j** NASPM inhibitory effect. All data are reported as mean and SEM. Results obtained from control mice are displayed in blue (ctrl) whereas data obtained from trained mice (train) are shown in orange. Filled symbols represent data recorded from Vgut2-positive neurons and open symbols are data recorded from Vglut2-negative cells

plastic changes following the acquisition and training of a complex forelimb movement.

**Vglut2- cells are necessary for fine motor control**. As a final step, to test the necessity of Vglut2-positive RN neurons in skilled motor behaviour, we used a chemo-genetic reversible approach injecting Vglut2-CRE mice with a viral vector coding the expression of inhibitory DREADDs (AAV1-EF1α-DIO-hM4D (Gi)-mCherry) to suppress the activity of RN Vglut2-positive cells during the SGT and the SPRT (Fig. 7a–c). The efficiency of the DREADDs was first assessed in vitro. The input-output curve had a leftward shift in the presence of clozapine N-oxide (CNO), indicating a hyperpolarization of the recorded neurons (Fig. 7d).

As expected, the failure rate in the SGT was dramatically increased in the presence of CNO compared to the baseline condition and following saline injection (Fig. 7e, f). Performance in the SPRT was also impaired in the presence of CNO (Fig. 7g–j). Notably, we observed no differences in spontaneous locomotion between saline or CNO injection in these mice (Supplementary Fig. 5). Injection of CNO in Vglut2-CRE-negative mice that did not express DREADDs left their performance unaffected (Fig. 7k)

confirming that the drop in motor accuracy was due to the inhibition of Vglut2-positive RN cells and not from ectopic effects of CNO. While chemogenetic inhibition, which does not fully prevent neurons from firing (Fig. 7d), did not recapitulate fully the impairment induced by complete ablation; it was sufficient to impair motor performance. It is conceivable that partial inhibition is not strong enough to reproduce the effect of the full RN lesion.

## Discussion
The role of the RN in fine motor skills has been studied in the past, however, how specific neuronal cell types encode the reaching movement has not been investigated so far.

We used a novel brain lesion approach to provide evidence for the RN's necessity in reaching and grasping skills. Light-mediated lesion revealed to be very consistent with a highly reproducible application, leading to cellular death restricted to the RN and an efficiency reaching up to 90%. In this case, spontaneous loco-motion was unchanged, whereas a dramatic impairment in fine motor skills including reaching and grasping was observed. Such behavioural phenotypes appear to be reversible as the failure and

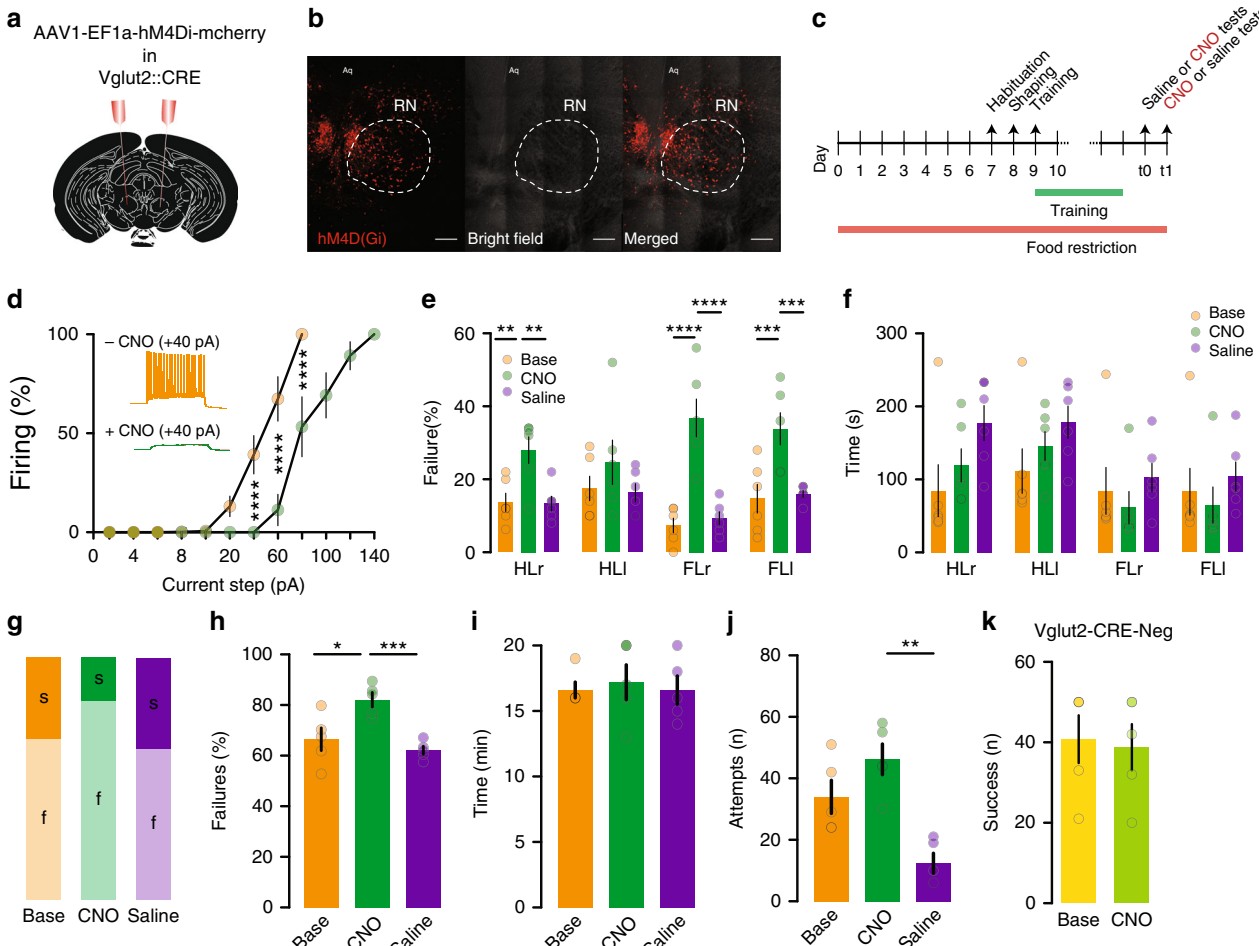

**Fig. 7** Vglut2-positive neurons are necessary for fine motor control. **a** Schematic of hM4D(Gi)-mcherry injection of Vglut2-CRE mice. **b** Viral expression (Scale bar 200 μm). **c** Timeline. **d** In vitro validation of hM4D(Gi), 2WRM-ANOVA $F_{(1,72)} = 62.58$, $P < 0.0001$ ($n = 7$ cells). **e** Failure rate in the EGT, 2WRM-ANOVA $F_{(2,40)} = 40.14$, $P < 0.0001$ ($n = 6$ mice) (HL: hindlimb, FL: forelimb, r: right, l: left). **f** Completion time. **g** Performance in the SPRT. **h** Failure rate, 1WRM-ANOVA $F_{(1.5,6)} = 28.23$, $P = 0.0012$. **i** Completion time. **j** Total attempts, 1WRM-ANOVA $F_{(1.5,6.1)} = 11.68$, $P = 0.01$. **k** Vglut2-CRE-negative mice success rate ($n = 6$ mice). All data are reported as mean and SEM. Baseline performance are depicted in orange or yellow, performance after CNO injection are shown in dark or light green and performance after saline injection are displayed in purple

time to complete both the SPRT and SGT one week after the lesion came back to pre-lesion performance levels. Considering that RN neurons were irreversibly lesioned, homoeostatic modifications occurring at parallel or complementary motor circuit pathways may account for the recovered reaching and grasping performance.

With the exception of the anatomical segregation of RNm and RNp, cellular identification and characterization of RN neurons has been largely neglected. Our FISH experiments revealed that Vgat expression is very limited in the RN suggesting that inhibitory control over RN cells might arise from other brain structures. In stark contrast, Vglut2 labelling was dense and overlapped with that of C1QL2, a previously reported marker for rubrospinal neurons[33]. Interestingly, Vglut2-positive neurons also co-expressed PV. Our in vitro electrophysiological recordings of photo-identified Vglut2-positive cells provided additional characterization; they sustain high firing rates which is in line with their expression of PV, a marker for fast-spiking neurons[39] and exhibit a high capacitance which reflect a large sized-cell. Our anatomical mapping investigation of Vglut2-expressing cells output targets labelled the ventromedial Thalamus, the Inferior Olive and the Spinal Cord. Based in the reported literature, the RNm projects to the spinal cord whereas the RNp projects to the

Inferior Olive[35,40]. However, it was also shown via WGA-HRP staining and mapping methods that RNp have a large caudal and lateral region that projects to the contralateral Spinal Cord and not the Inferior Olive[33,41]. Vglut2/PV positive RN neurons cannot therefore, be categorized as exclusively RNm or RNp, in line with studies reporting that both RNm and RNp are critical in skilled reaching when decomposing the movement into arpeggio, supination and supination[11].

The SPRT requires a learning phase where mice become more and more proficient at fine and precise movements. We show that Vglut2-expressing RN neurons undergo plastic changes specifically at excitatory synapses. The underlying molecular mechanism depends on a switch in AMPA receptor composition going from GluA2-lacking rectifying receptors before training to GluA2-containing linear AMPA receptors after training. Such plastic events and their underlying mechanisms have been intensely studied in other brain regions such as the cerebellar stellate cell synapses[31] or in the ventral tegmental area after drug exposure[42]. The lack of change in AMPA/NMDA ratio that we measured may be due to a concomitant change in NMDA receptors as previously reported in midbrain dopamine neurons[43]. Our data show that Vglut2-positive neurons undergo motor-learning dependent synaptic plasticity, complementing the

previously observed changes in synaptic contacts and in firing frequencies of RN cells after muscle conditioning[21,22] and provide additional knowledge on the characteristics of RN plasticity.

In addition to confirming that the RN in its entirety (light lesion experiments) is necessary for reaching and grasping skills, we further demonstrate that it does so via Vglut2-positive neurons. Indeed selective chemogenetic inhibition of RN Vglut2-expressing neurons recapitulated the impairment of fine movements in the SPRT and SGT and did not impact spontaneous locomotion. Our experimental observations provide additional knowledge into how the RN and precisely its Vglut2-expressing neuronal population produces reaching and grasping. However, a decomposed analysis of extension, supination, pronation and arpeggio has not been conducted in this present study.

Although the RN is necessary in reaching movement, its role is conditional to the activity of other brain areas both upstream and downstream from it. Our data shows that the RN sends direct projections to the spine, making it a crucial candidate in the execution of motor commands. Our loss of function experiments show that both after ablation or temporary chemo-genetic inhibition of the RN, motor function is impaired, showing the necessity of correct RN activity to maintain proper motor output. Despite severe impairments, mice were able to perform reaching movements. This suggests that the RN plays a role in the optimization or fine-tuning of movements and overall the motor function carried by other brain regions like the motor cortex would still be online even after RN ablation. The motor training-induced plasticity we show here, specifically in Vglut2 neurons, is a likely mechanism that allows the RN to fulfil this function.

Altogether, our data are exclusive in reporting that RN Vglut2-positive cells not only undergo a progressive change during the day to day acquisition and optimization of a complex motor movement, but they are also crucial for the proper execution of said movement once it has been perfected. Our study postulates the Vglut2-positive rubral population as an under-explored non-canonical motor centre exerting a high degree of control over complex tasks with a circuitry and underlying mechanisms providing fertile ground for future studies.

## Methods

**Subjects**. All animals and procedures were ethically approved by the Institutional Animal Care office of the University of Basel and the Cantonal Veterinary office under the License number 2742.

**Light ablation validation**. WT C57BL/6 mice were anaesthetized with isoflurane (Primal Healthcare), (5% induction and 1.5% during the rest of the procedure) and placed in a stereotaxic frame (WPI), skin and skull were thoroughly cleaned and sterilized with 70% ethanol and Betadine (Mundipharma) before drilling a hole above the RN. A single optic fibre (Thorlabs) was then implanted according to the stereotaxic coordinates +0.25 AP, 0.6 ML, −3.75DV from Lambda. The implant was then secured using dental cement (Lang Dental) and the animals were placed in a recovery cage with a heated floor.

One week after the surgical implant the animals were placed in an open field box where they were exposed to 50 mW 473 nm constant light for a total duration of 5 min. Twenty-four hours after the light exposure animals were given a lethal dose of pentobarbital and were intracardially perfused with PBS and 4% PFA. Brains were kept on 4% PFA overnight and then 60 μm slices were prepared using a vibrotome (Leica). The brain slices were then stained with Green fluorescent Neurotrace (Thermofisher) following the recommended protocol.

Confocal images of the lesioned area and its contralateral hemisphere were taken from three sections of each animal for a total of 12 imaged brain slices (n = 4 mice). A region of interest delineating the red nucleus was drawn on each hemisphere and the neurons present inside the ROI were counted for each slice and each hemisphere.

**Food restriction**. Animals were single housed and given ad libitum access to water and limited access to food. All animals were weighted daily and the amount of food placed in their caged was adjusted to maintain a body weight slightly above 85% of their starting weight.

**Single pellet reaching task**. Mice were food restricted for a week before being exposed to the SPRT chamber for 5 min. The following day the mice were placed in the chamber and given free access to 20 mg dustless chocolate flavoured pellets (Bioseb) until they consumed 50 pellets or 20 min of elapsed time were reached. This criterion was maintained as session end point for the rest of the training and testing sessions. After the shaping session (free access to pellets) all mice had a daily training session where a single pellet would be placed in front of the slit and progressively moved away from it to force the animals to reach the pellet with their paw and not the tongue. The daily training sessions were maintained until a stable performance was achieved across three consecutive days (usually after 7–12 days of training). After stability was achieved the mice received one last testing session where the performance was video recorded for post-hoc quantification of successful and failed reaching attempts.

**Suspended grid test**. A 30 × 30 cm grid with 2 mm diameter rods and 20 mm spacing was hanged with 4 anchoring points at 50 cm from the surface of a sound isolated behavioural box.

A video camera was placed under the grid to record the fore- and hind-paws of mice as they walked on the grid. Each mouse was manually placed in the centre of the grid and recorded for 3–5 min for post-hoc quantification of the accuracy of every step or attempt for each paw. A total of 50 steps per paw were quantified and categorized as a success attempts if the paw was correctly placed on the intended rod, or failure if the paw failed to contact any rod and fell through the spacing between rods.

**Open field test**. Mice were placed in the centre of a 30 × 45 cm box and could freely explore the area. Automatic video tracking software (Anymaze, Stoelting) was used to register the position and movement of the mice during a 6 min session.

The raw X;Y position coordinates were then exported and processed using a custom Matlab (Mathworks) script designed to decompose the locomotion trace into single bouts for quantification and analysis.

**In vitro electrophysiology**. Slices (200 μm) containing the RN were prepared using a vibratome (Leica) in ice-cold cutting solution (in mM: kynureic acid 3, NaHCO$_3$ 26.2, MgCl$_2$ 4.9, CaCl$_2$ 1.22, glucose 1.25 and sucrose 225). Slices were incubated in ACSF in mM: NaCl 119, NaHCO$_3$ 26.2, KCl 2.5, MgCl$_2$ (6H$_2$O) 1.3, NaHPO4 1, CaCl$_2$ 2.5, glucose 11, pH 7.3, at 31 °C before resting at RT. Slices were then transferred to the recording chamber, perfused with ACSF at a rate of 2 ml/min and oxygenated with 95%O$_2$ and 5%CO$_2$. Neurons were visualized with an IR camera on an Olympus U-TLUIR and whole-cell-patch clamp recordings (multi-clamp 700B amplifier) were made. The internal solution for plasticity experiments and recording of excitatory transmission contained in mM: KGluconate 140, KCl 5, creatine phosphate 10, MgCl$_2$ 2, Na$_2$ATP 4, Na$_3$GTP 0.3, EGTA 0.2 and HEPES 10, pH 7.3, osmolarity 300. NASPM was used at 100 μM. For IPSC recordings only cells with a capacitance above 30 pF were screened, the internal solution contained in mM: KGluconate 30, KCl 100, Creatine phosphate 10, MgCl$_2$ 4, Na$_2$ATP 3.4, Na$_3$GTP 0.1, EGTA 1.1 and HEPES 5, pH 7.3, osmolarity 289. In the case of electrophysiological profiling, the internal solution contained in mM: KGluconate 130, creatine phosphate 10, MgCl$_2$ 4, Na$_2$ATP 3.4, Na$_3$GTP 0.1, EGTA 1.1 and HEPES 5, pH 7.3, osmolarity 289.

In a subset of experiments identification of the recorded neurons was achieved using Vglut2-CRE mice stereotactically injected with AAV5-ef1α-DIO-ChR2-EYFP (UNC vector core) in the RN. A 500 ms 473 nm light pulse was delivered through the optical components of the recording apparatus and the presence of a desensitized current was used to identify Vglut2-positive cells.

**Fluorescent in situ hybridization**. Brains were quickly dissected and fresh frozen in dry ice and stored overnight at −80 °C. 15 μm slices were then prepared with a cryostat (Leica) and mounted on super frost plus charged slides (Thermofisher). Slices were then fixed, dehydrated, digested, hybridized and developed following the standard procedures (Advanced Cell Diagnostics) for a multiplex fluorescent in situ hybridization protocol. Confocal images were taken using a Zeiss LSM700 microscope, processed with FIJI (imagej).

**Anterograde tracing**. Vglut2-CRE mice were stereotactically injected with AAV5 phSyn1(S)-FLEX-tdTomato-T2A-SypEGFP-WPRE in the RN. Six weeks post surgery mice were intracardially perfused with PBS and 4% PFA, the brains and spinal cords were carefully dissected. After 3 days in 30% sucrose for cryoprotection, 60 μm slices of brain and spinal cord were prepared for confocal microscopy imaging.

**Chemogenetic experiments**. Vglut2-CRE mice were stereotactically injected with AAV1-EF1α-DIO-hM4D(Gi)-mCherry in the RN. After food restriction, training and fulfilment of the stable performance criteria, mice were then tested in the behavioural assays first to establish a baseline performance. On subsequent days the mice received an i.p injection of saline or CNO (2 mg/kg) 1 h prior to testing. The order of injection between saline and CNO was counterbalanced across animals.

A control group of Vglut2-CRE-negative mice underwent the same procedure and was used to exclude any effect of CNO alone.

**Statistical analysis**. Paired or unpaired Student's *t*-tests analysis (two-tailed) were performed where applicable. In the experiments where more than two conditions were evaluated, a one-way, two-way or repeated measures-ANOVA (1WRM-ANOVA or 2WRM-ANOVA) was performed where applicable, followed by multiple comparisons testing using the Tukey correction method. Post-hoc power analysis was performed for behavioural experiments. $*p < 0.05$, $**p < 0.01$, $***p < 0.001$, $****p < 0.0001$.

**Reporting summary**. Further information on research design is available in the Nature Research Reporting Summary linked to this article.

## Data availability
The datasets generated during and/or analysed during the current study are available from the corresponding author on reasonable request.

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

## Acknowledgements
We thank the past and present members of the Tan Lab for constructive input, the Biozentrum Imaging Facility, the G. Keller's laboratory for providing viral vectors. This work was supported by the Swiss National Science Foundation (PPOOP3_150683 and BSSGIO_155830) and the Biozentrum.

## Author contributions
G.R. and K.R.T designed and performed the experiments and analysed the data. M.C. helped with performing the SPRT and SGT.

**Additional information**

**Competing interests:** The authors declare no competing interests.

