## [Peer Review File · Nature Communications]

Reviewers' comments:

Reviewer #1 (Remarks to the Author):

It is stimulating to notice that some people are interested in the putative role of the red nucleus. For example, this important motor nucleus is even not mentioned in Kandel's Principles of Neural Sciences! Although not mentioned by the authors in their Introduction section, the presence of a definite RN and of the corresponding rubrospinal tract has been consistently related to the presence of limbs, as in many terrestrial vertebrates, and or limb-like, analogous structures as in rays (see, for example, ten Donkelaar, *Behav. Brain Res.*, 1988; Gruber and Gould, *Neuroanat.*, 2010). Nevertheless, in animals like rats, mice, and, obviously primates the red nucleus receives dense projections from the sensorimotor cortex and from deep cerebellar (interpositus and dentate) nuclei (Pong et al., *Brain Res Rev.*, 2008; Miller and Gibson, *Encyclopedia of Neuroscience*, v. 8, p. 55-62, 2009). This anatomical and hodological details should be considering when dealing with the red nucleus. Besides, I have some comments and suggestions aimed to improve the final presentation of this study.

- It is not possible to identify the proper location of the neural population co-expressing Vglut2, PV, and C1Q12. Authors claim that this population corresponds to the magnocellular subdivision of the red nucleus. This point should be supported with additional histological illustrations. For example, since magnocellular neurons projects to the spinal cord it should be rather easily to label them using a retrograde tracer. Besides, it would also be necessary to record the activity of those projecting neurons in alert but restrained mice to show that their firing are not only related to the activity of large body muscles, but also to the specific activity of hand muscles. Finally, those neurons could be identified by their antidromic activation from the rubrospinal track.

- I still have a question regarding the putative role of rubral neurons located in the parvocellular part of the red nucleus. These neurons are evolutionary more related to fine motor responses and are mostly related with dentate feedback projections to the thalamus and the sensorimotor and prefrontal cortex. In my opinion, those neurons should be better candidates to play the role ascribed here to magnocellular neurons.

- Grasping involve the activity of many muscles, from those involved in postural and arm adjustments to those involve in the fine manipulation of small and delicate objects. Perhaps, authors should use more specific tests aimed to separate finger movements from the rest of motor activities. This could be achieved recording the EMG of those muscles in simultaneity with the unitary activity of identified red nucleus cells.

- Since the seminal studies of Tsukahara (*Ann Rev Neurosci.*, 1981) it is commonly accepted that the red nucleus plays an integrative role of motor commands arriving to it from the cerebral cortex and from the cerebellar nuclei. But the point is whether red nucleus cells are the one in charge in the acquisition and storage of new motor abilities or whether they play a secondary role. This point is not clearly demonstrating here, in absence of the respective roles of cortical and cerebellar neurons projecting onto it (see for example, Pacheco-Calderón et al., *J. Neurosci.*, 2012). For example, Hasan et al. (*Nature Comm.*, 2013) have shown in mice the involvement of motor cortex neurons in the acquisition of both classical and instrumental conditioning tasks.

Minor comments

- Pag. 3, line 7. A video should be provided illustrating grasping movements of the experimental animals.

- It is difficult to see what is illustrated in Figure 1b. Please show some enlarged photomicrographs.

- Fig. 2a, b. Some retrograde labelling experiments should be performed to convincingly demonstrate that those neurons are projecting to the spinal cord, to the inferior olive or to the sensorimotor cortex.

- Page 7, 2nd line from bottom. Magnocellular cells are part of intrinsic rubral circuits. They are projecting neurons.

Reviewer #2 (Remarks to the Author):

The manuscript Rizzi et al., titled "Excitatory rubral cells encode the acquisition of novel complex motor tasks" showed that the change in the AMPA receptor composition in VGLUT2-positive neurons in red nucleus (RN) may be involved in the skilled motor behavior. Although the finding of the potential role of VGLUT2+ RN neurons in motor learning is interesting, several issues in the interpretation and data quality are concerning.

Major concerns:

1. Overall, most of data sets provided here are not clear. The SGT and SPRT performance data are so variable and the size of samples is small. Authors should do the power analysis to check whether enough sample data are provided to make statistical conclusion.

2. For chemogenetic manipulations, the interpretation of data is not clear. For example, the lesion study showed the reduced attempts in SPRT test (Fig. 1o), but chemogenetic silencing VGLUT2 neurons showed increased or no change attempts in SPRT (Fig. 3q). It is possible that VGLUT2 silencing is not enough to recapitulate the lesion study. Authors should address this issue.

3. The RN is important region for several motor function. It is possible that chemogenetic silencing of VGLUT2+ neurons alone can induce change in basal movement behaviors, including locomotion, gait and so on.

4. The synaptic mechanism for the change in AMPA receptor composition after training is interesting. However, mechanism is not clear. If this is the training induced LTP like mechanism, it is more likely to see increase in AMPA/NMDA ratio. However, there is no change in AMPA/NMDA ratio, while strong rectification of AMPA current in trained animals. Is it due to the increase in the GluR2-lacking receptors without affecting total AMPA current compared to NMDA current? Explanation is needed for possible mechanism. At the same time, the rectification data alone cannot clearly demonstrate the change in the AMPA receptor composition, the experiment with subunit specific antagonist should be performed.

5. Although the change in AMPA receptor composition after the training is interesting, this change alone cannot explain the plasticity induced by the training. Authors should check whether there is any change in the inhibitory components too. Furthermore, it should be very informative and strengthen the conclusion, if authors can do in vivo recording of VGLUT2 neurons during and after the training.

6. The description of data in Figure 3r is not clear. Did authors express hM4D channel in non-VGLUT2 neurons in RN? How? If author didn't specifically manipulate VGLUT2-negative neurons in RN, it is no different than control saline group. Then, unlike authors' comments, it is still possible that non-VGLUT2 neurons can participate in motor learning and skilled behaviors.

7. Further characterization including the projection of VGLUT2+ RN neurons will provide more information on this neural population.

Minor concerns:

1. The flow of manuscript is somewhat confusing especially for figure 1. The order of figures is not matched to the description in manuscript.
2. The font in some figures are too small, it will not be easy to read when figures are reduced to fit in the final reprint.

Reviewer #1 (Remarks to the Author):

It is stimulating to notice that some people are interested in the putative role of the red nucleus. For example, this important motor nucleus is even not mentioned in Kandel's Principles of Neural Sciences! Although not mentioned by the authors in their Introduction section, the presence of a definite RN and of the corresponding rubrospinal tract has been consistently related to the presence of limbs, as in many terrestrial vertebrates, and or limb-like, analogous structures as in rays (see, for example, ten Donkelaar, Behav. Brain Res., 1988; Gruber and Gould, Neuroanat., 2010). Nevertheless, in animals like rats, mice, and, obviously primates the red nucleus receives dense projections from the sensorimotor cortex and from deep cerebellar (interpositus and dentate) nuclei (Pong et al., Brain Res Rev., 2008; Miller and Gibson, Encyclopedia of Neuroscience, v. 8, p. 55-62, 2009). This anatomical and hodological details should be considering when dealing with the red nucleus.

Besides, I have some comments and suggestions aimed to improve the final presentation of this study.

- It is not possible to identify the proper location of the neural population co-expressing Vglut2, PV, and C1Q12. Authors claim that this population corresponds to the magnocellular subdivision of the red nucleus. This point should be supported with additional histological illustrations. For example, since magnocellular neurons projects to the spinal cord it should be rather easily to label them using a retrograde tracer. Besides, it would also be necessary to record the activity of those projecting neurons in alert but restrained mice to show that their firing are not only related to the activity of large body muscles, but also to the specific activity of hand muscles. Finally, those neurons could be identified by their antidromic activation from the rubrospinal track.

We agree with Reviewer 1 in that it is difficult to unequivocally identify RN cells belonging to either the magnocellular or parvocellular sub-divisions. With this study we followed an unbiased approach without the intent of specifically focusing on either magno- or parvo-, rather we followed the evidence and highlight the importance of Vglut2-positive cells in fine motor skills. We recognize that our Vglut2 oriented data is not sufficient to make indisputable claims about Vglut2 being an exclusive marker for magno- or parvo-subdivisions of the RN. For this reason we have toned down these claims in the manuscript.

We have however further characterized our population and we now provide histological data illustrating the output of Vglut2 RN neurons. We have expressed a GFP tagged synaptophysin specifically in Vglut2-positive neurons and tracked the labeled fibers.

This data is now reported in Figure 2q-t.

We are also now providing a low-magnification confocal image of one of the FISH experiment. This allows to better appreciate where our neurons of interest (Vgut2-positive) are located. This data is now reported in Sup. Figure 2.

- I still have a question regarding the putative role of rubral neurons located in the parvocellular part of the red nucleus. These neurons are evolutionary more related to fine motor responses and are mostly related with dentate feedback projections to the thalamus and the sensorimotor and prefrontal cortex. In my opinion, those neurons should be better candidates to play the role ascribed here to magnocellular neurons.

We observe strong innervation in thalamic areas as well as the inferior olive and the spinal cord. This suggests that while Vglut2 represents a valid functional marker to identify and manipulate cells involved in motor skill learning, we do not believe it is an anatomical marker that represents exclusively magno- or parvo- belonging cells.

- Grasping involve the activity of many muscles, from those involved in postural and arm adjustments to those involve in the fine manipulation of small and delicate objects. Perhaps, authors should use more specific tests aimed to separate finger movements from the rest of motor activities. This could be achieved recording the EMG of those muscles in simultaneity with the unitary activity of identified red nucleus cells.

Although we find this comment and suggested experiments very interesting, we would like the scope and novelty of our data set to lie in the identification of a sub-population that undergoes synaptic plasticity after motor training. Substantial previous literature has taken this approach (*in vivo recordings and EMG studies*: e.g., Cheney and Fetz 1988, Mewes and Cheney 1991, Miller et al., 199, Miller and Houk 1995, Belhaj-Saif et al., 1998, Miller and Sinkjaer 1998, van Kan and McCurdy 2001, *in vivo recordings studies*: e.g., Kohlerman et al., 1982, Gibson et al., 1985, Hermer-Vazquez et al., 2004) so we would like to branch away from this approach and provide a refreshing take on RN function that we would like to report in this initial study.

- Since the seminal studies of Tsukahara (Ann Rev Neurosci., 1981) it is commonly accepted that the red nucleus plays an integrative role of motor commands arriving to it from the cerebral cortex and from the cerebellar nuclei. But the point is whether red nucleus cells are the one in charge in the acquisition and storage of new motor abilities or whether they play a secondary role. This point is not clearly demonstrating here, in absence of the respective roles of cortical and cerebellar neurons projecting onto it (see for example, Pacheco-Calderón et al., J. Neurosci., 2012). For example, Hasan et al. (Nature Comm., 2013) have shown in mice the involvement of motor cortex neurons in the acquisition of both classical and instrumental conditioning tasks.

We agree with Reviewer 1 in that the role of the RN in motor function is conditional to the activity of other brain areas both upstream and downstream from it. Our data shows that the RN sends direct projections to the spine, making it a crucial candidate in the execution of motor commands. Our loss of function experiments show that both after ablation or temporary inhibition of the RN the motor function is impaired, showing the necessity of correct RN activity to maintain proper motor output. The animals however were still able to perform the reaching task albeit with strongly decreased efficiency. This suggests a more secondary role the RN plays in the optimization or fine tuning of movements, but overall motor function carried by other brain regions like the motor cortex would still be online even after RN ablation. The motor training induced plasticity we show here, specifically in Vglut2 neurons, is a likely mechanism that allows the RN to fulfill this function.

Minor comments

- Pag. 3, line 7. A video should be provided illustrating grasping movements of the experimental animals.

We are now providing video examples of all the behavioral paradigms we used.

- It is difficult to see what is illustrated in Figure 1b. Please show some enlarged photomicrographs.
- We would very much like to enlarge Figure 1b. Unfortunately space restriction for a short communication format does not allow us to do so. We however have modified the confocal image for more clarity.

- Fig. 2a, b. Some retrograde labelling experiments should be performed to convincingly demonstrate that those neurons are projecting to the spinal cord, to the inferior olive or to the sensorimotor cortex.

We have used a CRE-dependent GFP tagged version of synaptophysin to map the outputs of Vglut2+ neurons and we observe innervation in thalamic, brainstem and spinal cord regions. This data is now reported in the new Figure 2q-t.

- Page 7, 2nd line from bottom. Magnocellular cells are part of intrinsic rubral circuits. They are projecting neurons.

Unfortunately, we could not identify what Reviewer 1 was referring to with this comment, however there is no more reference to magnocellular cells in the new version of the manuscript.

Reviewer #2 (Remarks to the Author):

The manuscript Rizzi et al., titled “Excitatory rubral cells encode the acquisition of novel complex motor tasks” showed that the change in the AMPA receptor composition in VGLUT2-positive neurons in red nucleus (RN) may be involved in the skilled motor behavior. Although the finding of the potential role of VGLUT2+ RN neurons in motor learning is interesting, several issues in the interpretation and data quality are concerning.

Major concerns:

1. Overall, most of data sets provided here are not clear. The SGT and SPRT performance data are so variable and the size of samples is small. Authors should do the power analysis to check whether enough sample data are provided to make statistical conclusion.

The variability in our data set and the sample size are both comparable with similar studies performed in mice that have to adhere to strict animal welfare regulations. We fully agree with Reviewer 2 in that an ideal higher number of mice used in each experiment would reduce the risk of statistical errors. Following the advice of Reviewer 2, we have performed a post-hoc power analysis using SPSS (reported below) and we believe overall the sample size we used was appropriate to illustrate the motor impairments caused by ablation or inhibition of RN.

Multivariate Tests		Value	F	Hypothesis df	Error df	Sig.	Partial Eta Squared	Noncent. Parameter	Observed Power
Fig. 1h	Pillai's trace	0.901	18.213a	2	4	0.01	0.901	36.427	0.94
	Wilks' lambda	0.099	18.213a	2	4	0.01	0.901	36.427	0.94
	Hotelling's trace	9.107	18.213a	2	4	0.01	0.901	36.427	0.94
	Roy's largest root	9.107	18.213a	2	4	0.01	0.901	36.427	0.94
Multivariate Tests		Value	F	Hypothesis df	Error df	Sig.	Partial Eta Squared	Noncent. Parameter	Observed Power
Fig. 1i	Pillai's trace	0.739	5.654a	2	4	0.07	0.739	11.308	0.52
	Wilks' lambda	0.261	5.654a	2	4	0.07	0.739	11.308	0.52
	Hotelling's trace	2.827	5.654a	2	4	0.07	0.739	11.308	0.52
	Roy's largest root	2.827	5.654a	2	4	0.07	0.739	11.308	0.52
Multivariate Tests		Value	F	Hypothesis df	Error df	Sig.	Partial Eta Squared	Noncent. Parameter	Observed Power
Fig. 1m	Pillai's trace	0.859	12.149a	2	4	0.02	0.859	24.299	0.83
	Wilks' lambda	0.141	12.149a	2	4	0.02	0.859	24.299	0.83
	Hotelling's trace	6.075	12.149a	2	4	0.02	0.859	24.299	0.83
	Roy's largest root	6.075	12.149a	2	4	0.02	0.859	24.299	0.83
Multivariate Tests		Value	F	Hypothesis df	Error df	Sig.	Partial Eta Squared	Noncent. Parameter	Observed Power
Fig. 1n	Pillai's trace	0.897	17.380a	2	4	0.01	0.897	34.761	0.93
	Wilks' lambda	0.103	17.380a	2	4	0.01	0.897	34.761	0.93
	Hotelling's trace	8.69	17.380a	2	4	0.01	0.897	34.761	0.93
	Roy's largest root	8.69	17.380a	2	4	0.01	0.897	34.761	0.93
Multivariate Tests		Value	F	Hypothesis df	Error df	Sig.	Partial Eta Squared	Noncent. Parameter	Observed Power
Fig. 1o	Pillai's trace	0.882	14.923a	2	4	0.01	0.882	29.846	0.90
	Wilks' lambda	0.118	14.923a	2	4	0.01	0.882	29.846	0.90
	Hotelling's trace	7.461	14.923a	2	4	0.01	0.882	29.846	0.90
	Roy's largest root	7.461	14.923a	2	4	0.01	0.882	29.846	0.90
Multivariate Tests		Value	F	Hypothesis df	Error df	Sig.	Partial Eta Squared	Noncent. Parameter	Observed Power
Fig. 3o	Pillai's trace	0.83	9.785a	2	4	0.03	0.83	19.57	0.75
	Wilks' lambda	0.17	9.785a	2	4	0.03	0.83	19.57	0.75
	Hotelling's trace	4.892	9.785a	2	4	0.03	0.83	19.57	0.75
	Roy's largest root	4.892	9.785a	2	4	0.03	0.83	19.57	0.75
Multivariate Tests		Value	F	Hypothesis df	Error df	Sig.	Partial Eta Squared	Noncent. Parameter	Observed Power
Fig. 3r	Pillai's trace	0.966	42.365a	2	3	0.01	0.966	84.731	0.99
	Wilks' lambda	0.034	42.365a	2	3	0.01	0.966	84.731	0.99
	Hotelling's trace	28.244	42.365a	2	3	0.01	0.966	84.731	0.99
	Roy's largest root	28.244	42.365a	2	3	0.01	0.966	84.731	0.99
Multivariate Tests		Value	F	Hypothesis df	Error df	Sig.	Partial Eta Squared	Noncent. Parameter	Observed Power
Fig. 3t	Pillai's trace	0.928	19.305a	2	3	0.02	0.928	38.611	0.87
	Wilks' lambda	0.072	19.305a	2	3	0.02	0.928	38.611	0.87
	Hotelling's trace	12.87	19.305a	2	3	0.02	0.928	38.611	0.87
	Roy's largest root	12.87	19.305a	2	3	0.02	0.928	38.611	0.87

2. For chemogenetic manipulations, the interpretation of data is not clear. For example, the lesion study showed the reduced attempts in SPRT test (Fig. 1o), but chemogenetic silencing VGLUT2 neurons showed increased or no change attempts in SPRT (Fig. 3q). It is possible that VGLUT2 silencing is not enough to recapitulate the lesion study. Authors should address this issue.

We agree with Reviewer 2 and we now highlight in the manuscript that while chemogenetic inhibition did not recapitulate fully the impairment induced by complete ablation, it was sufficient to impair motor performance. It is indeed conceivable that partial inhibition is not strong enough to reproduce the effect of a full brain lesion.

We now address this observation in the manuscript.

3. The RN is important region for several motor function. It is possible that chemogenetic silencing of VGLUT2+ neurons alone can induce change in basal movement behaviors, including locomotion, gait and so on.

We have now tested the effect of chemogenetic silencing of Vglut2+ neurons on spontaneous locomotion and we observed no significant changes. These data are now reported in the new Supplementary Figure 5.

4. The synaptic mechanism for the change in AMPA receptor composition after training is interesting. However, mechanism is not clear. If this is the training induced LTP like mechanism, it is more likely to see increase in AMPA/NMDA ratio. However, there is no change in AMPA/NMDA ratio, while strong rectification of AMPA current in trained animals. Is it due to the increase in the GluR2-lacking receptors without affecting total AMPA current compared to NMDA current? Explanation is needed for possible mechanism. At the same time, the rectification data alone cannot clearly demonstrate the change in the AMPA receptor composition, the experiment with subunit specific antagonist should be performed.

Following Reviewer 2's suggestion we now provide experimental evidence demonstration the switch and composition of AMPAR after training. A new batch of mice were food-restricted, half of them underwent training in the SPRT and slice electrophysiology was performed right after the test. AMPA-mediated transmission was recorded at -60, 0 and +30 mV before and after bath application of NASPM a selective GluA2-lacking AMPA receptor antagonist. In control mice, where synapses are rectifying, NASPM consistently reduced the amplitude of the EPSCs recorded at negative potentials, confirming the presence of GluA2-lacking AMPA receptors. The calculated rectification index decreased towards linearity. In trained mice, where synapses are linear, NASPM had no effect, confirming the absence of GluA2-lacking AMPA receptors.

These new data are now reported in the new Figure 3h-j.

5. Although the change in AMPA receptor composition after the training is interesting, this change alone cannot explain the plasticity induced by the training. Authors should check whether there is any change in the inhibitory components too. Furthermore, it should be very informative and strengthen the conclusion, if authors can do in vivo recording of VGLUT2 neurons during and after the training.

1- We have now examined the inhibitory component, we measured both the electrically evoked IPSC paired pulse ratio (PPR) and the miniature activity (amplitude and frequency). We observed no change in these parameters. This data is now reported in the new Supplementary Figure 4.

2- Unfortunately, we do not have the technical ability to do in vivo recordings at the moment and we hope the new version of the manuscript (as a short communication) and our finding that are new to the field will be estimated satisfying.

6. The description of data in Figure 3r is not clear. Did authors express hM4D channel in non-VGLUT2 neurons in RN? How? If author didn't specifically manipulate VGLUT2-negative neurons in RN, it is no different than control saline group. Then, unlike authors' comments, it is still possible that non-VGLUT2 neurons can participate in motor learning and skilled behaviors.

We apologize if the information was not clearly stated in the manuscript. The data presented in the old Figure 3r (now as the new Figure 3u) is a control experiment to verify that CNO alone, in the absence of DREADD expression, does not have an effect on motor performance. WT littermate of Vglut2-CRE-negative mice were handled in parallel to the Vglut2-CRE-positive mice, specifically they received both the viral vector (but do not have the ability to recombine and induce hM4D expression) and CNO. This preparation provides evidence that CNO per se is not active in the brain, or at least it does not produce motor impairments in our context.

7. Further characterization including the projection of VGLUT2+ RN neurons will provide more information on this neural population.

We have used a CRE-dependent GFP tagged version of synaptophysin to map the outputs of Vglut2+ neurons and we observe innervation in thalamic, brainstem and spinal cord regions. This data is now reported in the new Figure 2q-t.

Minor concerns:

1. The flow of manuscript is somewhat confusing especially for figure 1. The order of figures is not matched to the description in manuscript. We have corrected the manuscript for this confusion.
2. The font in some figures are too small, it will not be easy to read when figures are reduced to fit in the final reprint. We have modified the font size.

REVIEWERS' COMMENTS:

Reviewer #1 (Remarks to the Author):

Authors has made a noticeable effort to address all of my queries and suggestions. I am already satisfied by their answers and I have no further questions. Perhaps, if there is space for this they should comment on the functional descriptions regarding the role of identified rubral neurons in behaving animals as indicated in my previous evaluation.

Reviewer #2 (Remarks to the Author):

The revised manuscript Rizzi et al., titled "Excitatory rubral cells encode the acquisition of novel complex motor tasks" addressed most of previous concern. Especially, authors provided results showing that excitatory, not inhibitory synaptic changes is important for training induced plasticity, which make it more clear on the synaptic mechanisms onto RN neurons. I think scientific findings in this manuscript is significant enough to be in Nature communication.

Still, I hope authors can provide more statistical detail, including posthoc analysis and so on. I also worry that all main three figures are very heavy, which make it difficult to read in publication. Some data should be moved to supplementary data.

Reviewer #1 (Remarks to the Author):

Authors has made a noticeable effort to address all of my queries and suggestions. I am already satisfied by their answers and I have no further questions. Perhaps, if there is space for this they should comment on the functional descriptions regarding the role of identified rubral neurons in behaving animals as indicated in my previous evaluation.

We appreciate the positive feedback from reviewer #1.

We have expanded the discussion section to elaborate on this point.

Reviewer #2 (Remarks to the Author):

The revised manuscript Rizzi et al., titled “Excitatory rubral cells encode the acquisition of novel complex motor tasks” addressed most of previous concern. Especially, authors provided results showing that excitatory, not inhibitory synaptic changes is important for training induced plasticity, which make it more clear on the synaptic mechanisms onto RN neurons. I think scientific findings in this manuscript is significant enough to be in Nature communication.

Still, I hope authors can provide more statistical detail, including posthoc analysis and so on. I also worry that all main three figures are very heavy, which make it difficult to read in publication. Some data should be moved to supplementary data.

We also appreciate the positive feedback of reviewer #2.

We have verified that all the statistical reporting is in line with the journal’s requirements for publication.

We have split the figures to facilitate the reading and flow of the manuscript.